# Wild-Grown and Cultivated *Glechoma hederacea* L.: Chemical Composition and Potential for Cultivation in Organic Farming Conditions

**DOI:** 10.3390/plants11060819

**Published:** 2022-03-18

**Authors:** Inga Sile, Valerija Krizhanovska, Ilva Nakurte, Ieva Mezaka, Laura Kalane, Jevgenijs Filipovs, Alekss Vecvanags, Osvalds Pugovics, Solveiga Grinberga, Maija Dambrova, Arta Kronberga

**Affiliations:** 1Latvian Institute of Organic Synthesis, 21 Aizkraukles Street, LV-1006 Riga, Latvia; valerija@osi.lv (V.K.); osvalds@osi.lv (O.P.); solveiga@osi.lv (S.G.); maija.dambrova@farm.osi.lv (M.D.); 2Department of Applied Pharmacy, Riga Stradins University, 16 Dzirciema Street, LV-1007 Riga, Latvia; 3Department of Pharmaceutical Chemistry, Riga Stradins University, 16 Dzirciema Street, LV-1007 Riga, Latvia; 4Institute for Environmental Solutions, “Lidlauks”, Priekuļi Parish, LV-4126 Cēsis, Latvia; ilva.nakurte@vri.lv (I.N.); ieva.mezaka@vri.lv (I.M.); laura.kalane@vri.lv (L.K.); jevgenijs.filipovs@videsinstituts.lv (J.F.); alekss.vecvanags@videsinstituts.lv (A.V.); 5Field and Forest, SIA, 2 Izstades Street, Priekuļi Parish, LV-4126 Cēsis, Latvia; arta.kronberga@fieldandforest.lv

**Keywords:** *Glechoma hederacea*, chemical composition, essential oils, cultivation, harvesting period

## Abstract

*Glechoma hederacea* L. is a medicinal plant that is known in traditional medicine for its anti-inflammatory, antibacterial, antiviral, and anticancer properties. This study evaluated the potential for commercial production of *G. hederacea* and compared the chemical composition and activity of 70% ethanol extracts and steam-distilled essential oils from wild-grown and cultivated *G. hederacea* collected in different harvesting periods. The main compounds identified in the 70% ethanol extracts were phenolic acids (chlorogenic and rosmarinic acids) and flavonoid O-glycosides. The essential oil varied in the three accessions in the range of 0.32–2.98 mL/kg^−1^ of dry weight. The extracts possessed potent antioxidant and anti-inflammatory properties in LPS-treated bone-marrow-derived macrophages. The results of flow cytometry show that extracts from different vegetation periods reduced the conversion of macrophages to the proinflammatory phenotype M1. The chemical composition varied the most with the different harvesting periods, and the most suitable periods were the flowering and vegetative phases for the polyphenolic compounds and essential oils, respectively. *G. hederacea* can be successfully grown under organic farming conditions, and cultivation does not significantly affect the chemical composition and biological activity compared to wild-grown plants.

## 1. Introduction

*Glechoma hederacea* L., or ground-ivy, is a medicinal plant that belongs to the Lamiaceae family, is native to Eurasia, and is widely distributed throughout the territory of Latvia in forests, shrubs, and roadsides [1]. The aerial parts of *G. hederacea* have been used in traditional medicine for centuries to treat colds, asthma, bronchitis, gastric diseases, diabetes, and inflammation [2,3]. *G. hederacea* provides anti-inflammatory, analgesic, antibacterial, antiviral, anticancer, diuretic, and antioxidant properties [4,5,6]. Its potential use in pharmacy applications [7] and as a food additive [8] has been studied. An in vivo study of *G. hederacea* hot water extract in a rat model indicated protective effects against cholestatic liver injuries [9]. The antioxidant activity of the hot water extract is significantly higher than that of vitamin C and Trolox [10]. The *G. hederacea* methanol extract exhibited an antibacterial inhibitory effect against 14 bacterial species [3]. The aqueous extract also reduced the expression of nuclear transcription factors that are related to inflammatory pathways and inflammatory cytokine production [9]. Both the ethanol and water extracts possessed depigmenting activity in vitro and in vivo [2,11]. The study by Zhou et al. [6] summarizes the various pharmacological uses of *G. hederacea* and its chemical components that exhibit clinically relevant biological activity. Industrial applications of its extracts to enrich food are possible since no toxicity was detected in the extracts at the tested concentrations (maximum concentration of 10 mg/mL) in the brine shrimp lethality assay [3].

*G. hederacea* contains a variety of secondary metabolites, including terpenoids, flavonoids, phenolic acids, alkaloids, and essential oils, which leads to its pharmacological effects [6]. Phenolic compounds such as phenolic acids, flavonoids, and their derivatives are the main biologically active substances in polar solvent extracts [6]. Phenolic compounds have aroused significant interest due to their antiaging, anti-inflammatory, antioxidant, and antiproliferative activities [12,13]. More than 30 phenolic compounds were found in *G. hederacea*’s aqueous, alcohol, and aqueous-alcohol extracts; detailed reports are available in [8,14,15,16,17,18,19].

Although studies show the commercial potential of *G. hederacea*, in Europe, raw ground-ivy material is still collected in the wild [20] and there are no commercially available varieties. Therefore, the domestication of well-adapted populations with a suitable chemical profile is necessary for growers to start the cultivation of ground-ivy. A controlled cultivation of *G. hederacea* would ensure that stable and high-quality raw materials are available for processing industries. Therefore, the aim of our study was to compare the chemical compositions and biological activities of 70% ethanol extracts and steam-distilled essential oils from wild-grown and cultivated *G. hederacea*, as well as the changes in their chemical composition after two subsequent years of cultivation. Aerial parts of three wild-grown *G. hederacea* accessions were harvested from different regions in Latvia. The collected wild *G. hederacea* plantlets were planted and grown under organic farming conditions and collected in different vegetation periods. Qualitative and quantitative analyses of the extracts were performed using the liquid chromatography–mass spectrometry and gas chromatography–mass spectrometry methods. The variation in antiradical activity, the total phenolic content, and the effects on the expression of CD80 and CD86 in bone-marrow-derived macrophages were examined.

## 2. Results and Discussion

### 2.1. Seasonal Variation in G. hederacea during Field Trials

Meteorological conditions differed in the growing seasons of 2019, 2020, and 2021 (Priekuli meteorological station data). The total precipitation from the 1st of April until the 31st of October was highest in 2019 (528 mm), but was lowest in the same period in 2021 (472 mm). The sum of the active temperatures (>5 °C) was 1787, 1692, and 1760 °C in 2019, 2020, and 2021, respectively. The growing seasons of 2019 and 2020 were characterized by an optimal temperature and moisture regime in May, June, and July. This scenario resulted in the better development of *G. hederacea*, and plant regrowth allowed for three harvests in 2019 and four in 2020. Due to a very hot and dry June and July in 2021, plant regrowth was reduced, and the aerial part of the ground-ivy could only be harvested twice. Cultivated local accessions significantly varied in plant height and dry herb yield. In spring, *G. hederacea* develops flowering shoots. According to Slade and Hutching [21], at the flowering stage, plants grow vertically (up to 60 cm). In this study, the average plant height was 17.6 cm (Table 1). The average height of the flowering shoots from the tested accessions varied between 13.5 cm (GH01) and 20.2 cm (GH03). The dry herb yield was highly correlated with the height of the flowering shoots; thus, the highest yield of 565.8 kg ha^−1^ was observed in accession GH03. After flowering, *G. hederacea* develops secondary aboveground stolons [22], and its aerial shoots only reach 10.1 cm in height on average. Due to the minimum clearance of the harvester being at 8 cm, none of the accessions have potential for commercial harvesting after the regrowth of vegetative shoots.

Under organic conditions, the ability to suppress weeds is an important trait because herbicides are prohibited from being used. Crop ground cover is negatively correlated with weed development; if the soil is covered with more crops, weed suppression is higher. The development intensity (soil coverage) of the tested local accessions significantly differed (Figure 1). Accession GH01 had the lowest plant height and yield and developed more slowly, which also contributed to its lower weed-suppression ability.

### 2.2. Qualitative Analysis of G. hederacea 70% Ethanol Extracts

Qualitative analysis of the 70% ethanol extracts of *G. hederacea* was performed by UHPLC-HRMS/MS with an IT-TOF mass analyzer and a DAD detector, and the chromatograms are shown in Appendix A. Identification was based on high-resolution mass spectra, fragmentation, analysis of the available reference substances, and the literature data. The tentative identification was also confirmed by scanning the corresponding aglycone (quercetin, luteolin, kaempferol, and apigenin) masses in MRM mode, followed by a parent scan using a tandem mass spectrometer. The results of the qualitative analysis are summarized in Table 2.

The main compounds identified in the 70% ethanol extracts of *G. hederacea* were phenolic acids (chlorogenic acid and rosmarinic acid) and flavonoid (quercetin, kaempferol, luteolin, and apigenin) O-glycosides, and a minor quantity of flavones (luteolin and apigenin) was also found. Several quercetin, luteolin, and apigenin glycosides in polar solvent extracts were previously discovered [15,18,22]. Small phenolic acids such as gallic, vanillic and ferulic acids described as being in *G. hederacea* [14,18] were not observed, nor were the alkaloids described by Kumarasamy et al. [23].

Three peaks (peaks **1**–**3**) showed a signal at *m*/*z* 355 [M+H]^+^ (C_16_H_18_O_9_), corresponding to chlorogenic acid and its isomers. Peak **2** was identified as chlorogenic acid by comparing its retention time with that of a reference compound. However, the retention time of the chlorogenic acid isomers (peaks **1** and **3**) did not correspond to the retention time of the available cryptochlorogenic acid reference compound. Peak **3** was tentatively identified as neochlorogenic acid, and the identification was based on the elution order of the chlorogenic acids in reversed-phase chromatography [24]. Peak **1** was tentatively identified as cis-chlorogenic acid. Chlorogenic acid was previously obtained in *G. hederacea* extracts [10,20]; in turn, chlorogenic acid isomers were reported for the first time.

Three peaks (peaks **8**, **9**, and **13**) showed a 162 Da (C_6_H_10_O_5_) fragment elimination with aglycone formation at *m*/*z* 303, 287, and 271 [M+H]^+^, which correspond to quercetin, luteolin, and apigenin, respectively. Peaks **8**, **9**, and **13** were identified as hyperoside (quercetin 3-O-galactoside), luteolin 7-O-glucoside, and apigenin 7-O-glucoside, respectively, by comparisons with the reference compounds. A similar fragmentation with 2C_6_H_10_O_5_ fragment elimination and aglycone formation at *m*/*z* 303 [M+H]^+^ resulted in peak **4**, which was tentatively identified as quercetin-3-O-diglucoside. Quercetin 3-O-diglucoside was not previously observed in *G. hederacea*.

Two intense peaks (peaks **5** and **7**) of quercetin diglycosides were detected, with equivalent precursor ions at *m*/*z* 611 [M+H]^+^ and different fragmentations (611 > 449 > 303 vs. 611 > 465 > 303). Peak **7** was identified as rutin (quercetin 3-O-rutinoside), which was confirmed by the reference compound, and peak **5** was tentatively identified as quercetin 3-O-galactosyl-rhamnoside. The identification is based on the elution order of glycosylated flavonoids in reversed-phase chromatography [25,26]. Quercetin 3-O-galactosyl-rhamnoside has not been previously reported in *G. hederacea*.

In addition to the flavonoids monoglycoside and diglycoside (peaks **4**, **5**, **7**–**9**, **12**, and **13**), acylated glycosides were identified in the extracts. Four peaks (peaks **6**, **10**, **11**, and **15**) of quercetin and kaempferol glycoconjugates showed a 306 Da (C_12_H_18_O_9_) fragment elimination. Similar fragmentation was observed for flavanol glycoconjugates [27] acylated with 3-hydroxy-3-methylglutaric acid. Peaks **6**, **10**, **11**, and **15** are tentatively identified as kaempferol and quercetin-acylated glycoconjugates. These four glycoconjugates were not previously observed in *G. hederacea*.

### 2.3. Quantitative Analysis of the Main Phytocomponents in 70% Ethanol Extracts of G. hederacea

To determine the effect of the cultivation and harvesting time on the chemical composition of the plant, 10 major polyphenolic compounds (rosmarinic acid, chlorogenic acid, caffeic acid, rutin, apigenin 7-O-glucoside, luteolin 7-O-glucoside, apigenin, luteolin, hyperoside, and kaempferol 3-O-rutinoside) were selected, and their quantitative analysis was performed. The results are summarized in Table 3 and Table 4. The major polyphenolic compound content in 70% ethanol extracts prepared from three accessions of wild-grown and cultivated plants harvested in the same growth phase (flowering shoot) and that in cultivated plants collected in different vegetation periods (May–September) were compared. Previously, Chou et al. [14] stated that rosmarinic acid, chlorogenic acid, caffeic acid, rutin, genistin, and ferulic acid were the most abundant phytochemicals in the hot water extract of *G. hederacea* and possessed potent antioxidant and anti-inflammatory properties. Our study is the first to provide data about quantitative analyses of apigenin 7-O-glucoside, luteolin 7-O-glucoside, apigenin, luteolin, hyperoside, and kaempferol 3-O-rutinoside in *G. hederacea*.

### 2.4. Comparison of the Content of the Main Phytocomponents in 70% Ethanol Extracts of G. hederacea Prepared from Wild-Grown and Cultivated Accessions

The content of 10 main polyphenolic compounds in 70% ethanol extracts prepared from the flowering shoots of wild and cultivated plant accessions was analyzed. The results obtained show that the contents of both kaempferol 3-O-rutinoside and rutin were significantly lower in the extracts prepared from two accessions (GH02 and GH03) of the cultivated plant. In contrast, the GH01 accession yielded a considerably higher content of these compounds in its extracts after the first year of cultivation. After the second year of cultivation, the kaempferol 3-O-rutinoside and rutin contents were slightly lower than those in the wild-grown plants. The content of other flavonoids in extracts prepared from wild-grown plants is similar to that yielded by cultivated accessions. However, after the second year of cultivation, a slight lowering in the concentration of the compounds was observed.

The content of two flavone derivatives, apigenin and luteolin, in extracts prepared from accession GH01 was higher after the first year of cultivation, while the other two accessions, GH02 and GH03, yielded concentrations similar to those obtained from the wild-grown plants. In the second year of cultivation, the apigenin and luteolin contents in all extracts were similar to those obtained from wild-grown plants.

The chlorogenic acid content in extracts prepared from cultivated flowering shoots in the first year was significantly higher than that in extracts prepared from wild-grown flowering shoots. However, in the second year of cultivation, the chlorogenic acid concentration in the extracts was similar to that of wild-grown plants. Although a similar trend was observed for rosmarinic acid, it was mainly due to the high content of this phenolic in extracts prepared from the GH01 accession after the first year of cultivation. In the second year of cultivation, the content of rosmarinic acid was similar in extracts prepared from all the accessions and reached the level of that in wild-grown plants. Cultivation did not affect the caffeic acid content in the extracts. The average content of the main phenolic compounds in wild-grown and cultivated plants is summarized in Table 3 and Table 4.

### 2.5. Comparison of the Main Phytocomponents’ Contents in the 70% Ethanol Extracts of G. hederacea Prepared from Cultivated Specimens That Were Harvested in Different Vegetation Periods

The content of flavone derivatives, apigenin, and luteolin was higher in the extracts prepared from flowering shoots. In extracts prepared from vegetative shoots harvested in July and September 2020, only trace amounts of apigenin were detected, and the luteolin level was below the detection limit in all samples except one. The extracts prepared from plants harvested in June 2020 showed a certain level of both compounds, although the level was considerably lower than in those prepared from flowering shoots. Neither apigenin nor luteolin were detected in extracts that were prepared from the vegetative shoots harvested in July 2021 (GH01, GH02, GH03), compared to the 97.0 µg/g and 39.5 µg/g of dry material from apigenin and luteolin, respectively, found in extracts prepared from flowering shoots harvested in June 2021. The content of both apigenin 7-O-glucoside and luteolin 7-O-glucoside in the extracts closely follows the pattern of apigenin and luteolin. Similarly, the concentration of kaempferol 3-O-rutinoside tended to be higher in extracts prepared from flowering shoots. However, the very low overall content of this compound in extracts prepared from cultivated accessions GH02 and GH03 makes the pattern less explicit compared to extracts prepared from accession GH01. The hyperoside content in extracts prepared from flowering shoots harvested in 2021 was significantly higher (*p* = 0.003) than that in vegetative shoots. In contrast, the results for extracts prepared from plant material harvested in 2020 scatter very much, so a statistically justified conclusion cannot be made (Table 4).

The caffeic acid content in extracts prepared from the plants harvested in July 2020 was significantly higher (*p* = 0.02) than that in the extracts of plant material harvested in June 2020, despite both extracts being prepared from vegetative shoots. Similar contents of caffeic acid (294 ± 78 vs. 319 ± 50) were found in extracts prepared from material harvested in September. In contrast, the extracts prepared from the flowering shoots contained significantly less (*p* = 0.02) caffeic acid. Surprisingly, other phenolic acids showed the opposite tendency. Significantly (*p* = 0.009) less rosmarinic acid was found in the extracts prepared from vegetative shoots harvested in 2021 (3218 ± 592 µg/g) compared to those prepared from flowering shoots (5105 ± 362), with chlorogenic acid levels of 1461 ± 363 µg/g vs. 3390 ± 461 µg/g (*p* = 0.005), respectively (Table 3).

*G. hederacea* was found to contain hyperoside, rutin, kaempferol 3-O-rutinoside, luteolin 7-O-glucoside, chlorogenic acid, rosmarinic acid, and caffeic acid in both the flowering and vegetative shoots. In contrast, luteolin, apigenin, and apigenin 7-O-glucoside were only found in the flowering shoots (in plants harvested in May and June). The results are summarized in Table 4. Vegetative shoots are relatively rich in caffeic acid, but flowering shoots are rich in luteolin, apigenin, apigenin 7-O-glucoside, and luteolin 7-O-glucoside (Figure 2).

### 2.6. Quantitative Analysis of the Main Components of the Essential Oil of G. hederacea

As part of the secondary metabolism of plants, essential oils play an important role in a wide range of biological activities. Aromatic and volatile essential oil products can be used as natural additives to reduce oxidation and prevent inflammation, although their chemical composition can be very variable due to their susceptibility to ecological, ontogenetic, climatic, postharvest, and intraspecies genetic factors [15]. The amount of essential oils and the composition of wild *G. hederacea* have been presented in several studies [15,28,29,30]; however, no information on its commercial propagation was found. Previous research on the chemical composition of the essential oil of *G. hederacea* from Lithuania and Serbia [28,29,30] showed that terpenes represented the main portion of the oil. The major constituents were mostly sesquiterpene hydrocarbons and constituents with a germacrene skeleton. Germacrene D dominated all oils from the Vilnius district [30], while palmitic and linoleic acids were the main oils of the Serbian *G. hederacea* oil, along with germacrene D [29].

Hydrodistillation of *G. hederacea* yielded a pale yellow-green, pleasant-smelling essential oil with a dry weight concentration varying in the range of 0.32 to 2.98 mL·kg^−1^ (0.03–0.3%) for the plant material obtained in different vegetation periods (Table 5). Previous studies of plant material collected during the flowering period reported a lower yield of essential oils [28,29,30]. We observed that the potential to produce essential oils is slightly higher just after the flowering period, when the plant is busy spreading widely throughout its aboveground runners. The year of cultivation, the phenological stage, and the accession had a significant effect on the essential oil content; therefore, these factors must be taken into account regarding the quality of oil production.

The chemical compositions of the essential oils were determined according to their retention time and the spectrometric electronic library (NIST). The identities of the oil constituents were established using GC retention indices (RIs). In total, 64 compounds were identified in the essential oil of *G. hederacea* (Appendix A). The main compounds identified in the oil were β-myrcene, β-ocimene, germacrene-D and germacrene-B, eucalyptol, and 1-octen-3-ol. The evaluation of the ten most dominant volatile compounds of the essential oils obtained from all accessions is summarized in Figure 3. According to the hierarchical clustering, the *G. hederacea* samples were clustered into three main groups. Group one contained relatively high amounts of β-myrcene, cis-β-ocimene, and 1-octen-3-ol. The samples in group one consisted of vegetative shoots of cultivated plants collected in 2019. Group two was constituted by high levels of germacene-D, germacene-B, and δ-guaiene essential oil compounds. Group three was rich in cumene, β-elemene, and bicyclogermacene. Groups two and three included both vegetative and flowering shoots over several seasons. Germacrene D was found to be one of the dominant components in *G. hederacea* samples analyzed in Lithuania, where germacrene D was detected at a rate of 20.7%. Our results show a similar trend in samples analyzed from North America (germacrene D, 19.4%) and Serbia (germacrene D, 7.3%), where one of the predominant chemical compounds is germacrene D [28,29]. Significant amounts of separated eucalyptol, also known as 1,8-cineole, should be noted among the dominant compounds. Eucalyptol is well known for its antiseptic and expectorant activity, as well as its antiviral activity [31]. Although the *G. hederacea* samples studied had a lower essential oil content than that of widely cultivated essential oil plants, such as chamomile, whose essential oil content exceeds 6 mL·kg^−1^ [32], peppermint at 8 mL·kg^−1^ [33], and salvia at 24 mL·kg^−1^ [34], it can be used as a potential source of niche essential oil rich in germacrene B, germacrene D, β-ocimene, eucalyptol, and β-myrcene.

### 2.7. Correlation of the Essential Oil Composition and Ethanol Extract Composition

The correlation of the variables can be determined based on their loading plots (Figure 4a). A small, large, or 90° angle implies positive correlation, negative correlation, or no correlation between two components, respectively. As indicated in Figure 4a, the contents of germacene D, germacene B, chlorogenic acid, rutin, bicyclogarmacene, δ-guaiene, and kaempferol 3-O-rutinoside were correlated. This cluster of components had a weak or no correlation with another group of mutually correlated compounds–luteolin 7-O-glucoside, luteolin, apigenin 7-O-glucoside, apigenin, and cumene. In contrast to the flowering samples grouped more on the right side of the X axis in Figure 4c, both of these groups corresponded to the grouping of the vegetative samples on the left side. The flowering samples had more beta-myrcene, cis-beta-ocimene, and caffeic acid than other samples. The samples that were from different locations (Figure 4d) and were collected in different years (Figure 4b) formed overlapping clusters, indicating no differentiation due to these factors. The chemical composition is stable across different growing seasons and different vegetative shoots within seasons. Different chemotypes of the flowering and vegetative shoots can be used for different end-user purposes.

### 2.8. Total Content of Phenolic Compounds, Flavonoids, and DPPH Free Radical Scavenging Activity of G. hederacea Extracts

The capacity of the *G. hederacea* extracts to scavenge stable DPPH radicals, and the total phenolic and flavonoid contents in the extracts of the cultivated accessions, were higher after harvest in the later stages of vegetative growth (July–September) (Table 6). Although the flowering phase is mentioned in the literature as the time for aerial part collection, scientific evidence to support this has never been published. Our results show that *G. hederacea* contains valuable substances and can also be harvested later in the growing season; vegetative shoots are also valuable and can be harvested several times from May to September. Our results are similar to those of the study by Varga et al. [20], in which the TPC and antioxidant capacity of the water extracts of various cultivated populations of *G. hederacea* in Hungary at different harvest times showed that the values of those collected during summer (July) were higher than those of early spring (April) or late fall (October). All extracts prepared from the cultivated samples showed higher activity than that of the wild samples. In the wild, accession GH03 showed the best results, but in the cultivation process in 2020 and 2021, the differences between the three accessions related to chemical composition were revealed, and GH03 showed slightly lower activity than GH01 or GH02. However, the antiradical activity and total phenolic and flavonoid contents in all cultivated samples during different vegetation periods were still slightly higher than those in wild-grown *G. hederacea* extracts, suggesting that the three accessions, from the perspective of chemical composition, have the potential for commercial cultivation.

### 2.9. The Effect of G. hederacea Extracts on Bone-Marrow-Derived Macrophage Polarization toward the M1 Phenotype

In the present study, we tested how *G. hederacea* extracts affect LPS- and IFN-γ-stimulated macrophage polarization of the proinflammatory phenotype M1, characterized by the presence of the surface markers CD80 and CD86. The number of M1-polarized macrophages increased four-fold compared to that in the untreated control (Figure 5). Accession GH01 was selected for further experiments because it showed the most promising results after chemical composition analysis. The flow cytometry results indicate that extracts from different vegetation periods and at a concentration of 500 µg/mL reduced macrophage conversion of the proinflammatory phenotype M1 by 13–56%. The results of the study show that the *G. hederacea* extract generated from the July 2020 accessions had the strongest effect on the polarization of M1 macrophages (Figure 5C). That the highest activity was seen in the extract from July 2020 correlates with the phytochemical composition data; this extract contained the highest amount of biologically active components. The ability to reduce the M1 macrophage population decreased as follows, in order from highest to lowest: July 2020 > September 2020 > May 2020 > June 2021 > May 2019 (wild-grown) > July 2021. In general, the *G. hederacea* extracts cultivated in 2020 showed the highest activity. According to the results of the MTT assay, the extract of *G. hederacea* was not toxic to BMDMs when applied for 24 h at concentrations ranging from 50 to 750 μg/mL (Appendix A).

Rosmarinic acid and caffeic acid are phenolic acids found in a variety of plants, especially those of the Lamiaceae family [35]. Rosmarinic, caffeic, and chlorogenic acids were the main compounds identified in the extracts of *G. hederacea;* therefore, the activity of these substances in M1 macrophages was also examined. Rosmarinic acid and caffeic acid significantly decreased the LPS-induced expression of double-positive CD80 and CD86 cells (Figure 5D). According to quantitative analyses, the rosmarinic acid content in the *G. hederacea* extracts was the highest among all the components. Furthermore, rosmarinic acid showed the highest ability to reduce the population of M1 macrophages compared to caffeic and chlorogenic acids. Rosmarinic acid at a concentration of 100 mM reduced the level of CD80- and CD86-positive cells by 29% compared with the LPS/IFN-γ control. Chou et al. [14] demonstrated that the hot water extract of *G. hederacea* inhibited NF-кB expression in LPS-stimulated RAW264.7 macrophages through the reduction of ROS levels and the downregulation of the expression of proinflammatory genes, which may be related to the presence of rosmarinic acid, chlorogenic acid, caffeic acid, rutin, genistin, and ferulic acid.

## 3. Materials and Methods

### 3.1. Plant Material

The aerial parts of three *Glechoma hederacea* habitats were collected for chemical analysis at the flowering stage during May 2019. The locations the wild accessions were obtained from are summarized in Appendix A. The vouchers were deposited at the Institute for Environmental Solutions (IES) in Latvia under codes GH01, GH02, and GH03.

### 3.2. Field Trials and Preparation of G. hederacea Extracts

The field trials with *G. hederacea* were set up in two stages. First, randomly selected vegetative shoots were collected from three accessions in the wild in April 2019 and planted in an organically certified experimental field at the IES (57°19′11.7′′ N 25°19′18.8′′ E, 115 m altitude). The plot size was 6 m^2^ (one replication), and the plant spacing was 30 × 75 cm. The aerial parts (vegetative shoots) were collected for chemical analysis after the plants were regrown on 1 July 2019 and 23 August 2019 (hereafter referred to as “cultivated”) by cutting shoots approximately 8 cm above the soil surface in all plots and taking an average sample of 150 g for chemical analysis. In 2020, aerial parts from the same plots were collected four times—on the 18 May (flowering shoots), 1 June, 21 July and 15 September (vegetative shoots).

Vegetative shoots were collected from the same plots on 23 July 2020, and a new trial was carried out with the aim of evaluating the suitability of accessions for commercial cultivation. The field assays were set up as a randomized complete block design in four replications. The plot size was 16.2 m^2^ (6 rows with 75 cm between rows). The shoots were cut into 5–15 long pieces before being embedded in the soil in rows.

In 2021, the plants’ regrowth was slower, and the aerial parts were collected on the 1st of June (flowering branches) and the 23 July (vegetative branches). The aerial parts of each plot were cut approximately 8 cm high, simulating mechanical harvesting, and were collected and weighed, and the yield of the fresh part was calculated. The aerial parts collected from *G. hederacea* were air dried at room temperature. After drying the weighed herbs, the yield of the dry herbs was calculated (with 12% moisture). Additionally, the percentage of soil covering the accessions was measured every 2–3 weeks during the vegetation seasons of 2020 and 2021 (Figure 6). A quadrocopter platform with vertical take-off and landing (VTOL) was used to collect a set of aerial images at a fixed flight altitude of 12 m over the experimental crop fields. The RGB sensor was mounted during the flights to obtain high-resolution image data. An automated Python code was developed to process the collected data. Soil coverage (%) by green plant segments was calculated for each separate plot and used for statistical analysis and visualization.

The dried herbs from all replications were mixed, and an average sample of 150 g was taken for chemical analysis. Powdered and dried samples of *G. hederacea* were macerated with 70% ethanol solution in water at 1:10 *w*/*v*. The prepared solutions were incubated for 7 days in a dark, cool place and were frequently shaken. Afterward, the material was pressed, and the remaining solid was squeezed to remove all the remaining solvent. The obtained solutions were clarified by decantation and centrifugation. Subsequently, the extracts were concentrated using a rotary evaporator. Finally, the solutions underwent lyophilization. The powder was labeled and stored in a refrigerator at −20 °C prior to further analysis.

In vitro and ex vivo experiments with *G. hederacea* extracts were carried out using lyophilized plant material, which was dissolved in distilled water.

### 3.3. Chemicals and Reagents

LiChroslov hypergrade acetonitrile and formic acid were purchased from Sigma-Aldrich (Schnelldorf, Germany), and water for UHPLC analysis was purified using a Milli-Q Plus system (Merck Millipore, Burlington, MA, USA). Reference compounds chlorogenic acid, rosmarinic acid, and caffeic acid were purchased from Sigma-Aldrich (Schnelldorf, Germany). All reference flavonoid compounds were acquired from PhytoLab (Vestenbergsgreuth, Germany). 3-(4,5-dimethylthiazol-2-yl)-2,5-diphenyltetrazolium bromide (MTT), 2,2-Diphenyl-1-picrylhydrazyl (DPPH), Folin-Ciocalteu reagent, sodium carbonate (Na_2_CO_3_), gallic acid, L-ascorbic acid, aluminium trichloride (AlCl_3_), quercetin, foetal bovine serum (FBS), Hank’s buffered saline solution (HBSS), and trypsin, lipopolysaccharide (LPS) were obtained from Sigma Aldrich. RPMI-1640 medium with Glutamax was produced by Gibco. Mouse monocyte-colony stimulating factor (M-CSF) and interferon-gamma (IFN-γ) were obtained from PeproTech (London, UK). FITC-conjugated anti-mouse F4/80, phycoerythrin (PE)-conjugated anti-mouse CD86, and biotin-conjugated anti-mouse CD80 antibodies were purchased from BioLegend (San Diego, CA, USA).

### 3.4. UHPLC-HRMS/MS Analysis

UHPLC-HRMS/MS analysis was performed on a Shimadzu LC–MS hybrid IT-TOF system combined with a Nexera X2 UHPLC system. The separation was carried out with an Acquity UPLC BEH C18 column (2.1 × 150 mm, 1.7 µm particle size) using gradient elution with mobile phases A (0.1% formic acid in water) and B (acetonitrile) at a flow rate of 0.4 mL/min. The gradient conditions were as follows: 2% B, 1 min—2% B; 4 min—5% B; 14 min—15% B; 36 min—50% B; 48 min—98% B; 55 min—98% B; 58 min—2% B; 60 min—2% B. The column oven was set at 30 °C, the autosampler was set at 10 °C, and the sample injection volume was 1 µL.

The following mass spectrometer operating parameters were utilized: electrospray ionization in positive and negative ionization modes, mass scan range (*m*/*z*)—from 120 to 1000; detector voltage—1.5 kV; nebulizing gas (N_2_) flow—1.5 mL/min; ion accumulation time—100 ms; CDL temperature—250 °C; collision gas—argon and collision energy was set at 50%. LC–MS data were processed by LabSolutions software. A diode array detector was used to record UV–Vis spectra over a range from 210 nm to 800 nm.

70% ethanol extracts of *G. hederacea* were injected into the chromatographic system without preliminary processing.

### 3.5. UHPLC–MS/MS Analysis of Flavonoids

Quantitative analysis of flavones and flavonoid glycosides was performed using a Xevo TQ-S micro (Waters) tandem mass spectrometer that operated in the positive electrospray ionization and multiple reaction monitoring (MRM) modes. The MRM parameters of each compound were optimized by infusion in a mass spectrometer and are detailed in Appendix A. Chromatographic separation was achieved on an Acquity BEH C18 column (2.1 × 100 mm, 1.7 µm, Waters) at a constant temperature of 30 °C using a Waters Acquity UPLC system with a gradient elution with mobile phases A (0.1% formic acid in water) and B (acetonitrile) at a flow rate of 0.4 mL/min. The gradient parameters were as follows: 5% B, 0.5 min—5% B; 8 min—98% B; 10 min—98% B; 11 min—5% B; 12 min—5% B. The autosampler was set at 10 °C, and the sample injection volume was 1 µL.

Before analysis, samples were diluted 10 or 100 times with reserpine solution (internal standard) (10 ng/mL) in 70% ethanol; for each sample, two measurements were taken. The concentration of individual components in the extracts was determined by a calibration curve (ranging from 50 ng/mL to 10 µg/mL for all analytes).

### 3.6. UHPLC–MS/MS Analysis of Phenolic Acids

Quantitative analysis of phenolic acids was performed using a Quattro Micro (Waters) tandem mass spectrometer that operated in the negative electrospray ionization mode. Multiple reaction monitoring (MRM) parameters are detailed in Appendix A. Chromatographic separation was achieved on an Acquity BEH C18 column (2.1 × 50 mm, 1.7 µm, Waters) at a constant temperature of 30 °C using a Waters Acquity UPLC system with a gradient elution with mobile phases A (0.1% formic acid in water) and B (acetonitrile) at a flow rate of 0.25 mL/min. The gradient parameters were as follows: 5% B, 2.5 min—98% B; 4.5 min—98% B; 4.7 min—5% B; 6 min—5% B. The autosampler was set at 10 °C, and the sample injection volume was 5 µL. Samples before analyses were diluted 10 or 200 times with 70% ethanol; for each sample, two measurements were taken. The calibration concentrations ranged from 50 ng/mL to 5 µg/mL for all analytes.

### 3.7. Determination of Essential Oil Content

Plant samples dried at room temperature (~25 °C) were used for analysis. Essential oils were prepared using a Clevenger-type hydrodistillation apparatus. Fifteen grams of powdered herbal drug were transferred to a 500 mL flask, distilled water was added as the distillation liquid, and 0.50 mL of xylene was added to a graduated tube. The distillation was carried out at a rate of 3–4 mL·min^−1^ for 3 h. The essential oil yield (mL·kg^−1^) was calculated based on the dried weight of the samples. The oil dissolved in the organic layer was separated and dried over anhydrous sodium sulfate to eliminate moisture. The samples were preserved in a sealed amber glass vial at 4 °C until GC–MS analysis. The samples were replicated 3 times.

### 3.8. GC–MS Analysis of Essential Oils

For the test solution, 100 µL of the essential oil sample was diluted with 900 µL of cyclohexane and was mixed. Analyses were performed on an Agilent Technologies 7820A gas chromatograph coupled to Agilent 5977B mass selective detector (MSD) equipment. A nonpolar HP-5 capillary column (60 m × 0.25 mm, 0.25 µm film thicknesses) was coated with 5% phenyl and 95% methyl polysiloxane. The carrier gas was helium (He) with a split ratio of 1:100, and a flow rate of 1.3 mL/min was applied. The volume of injection was 3 μL. The temperature program began at 70 °C for 3 min and then increased at a rate of 10 °C/min to 290 °C; finally, 290 °C was maintained for 10 min. The injector temperature was 270 °C. Mass spectra were recorded at 70 eV. The mass range was from *m*/*z* 30–550. The ion source temperature was maintained at 230 °C. The components were identified based on their retention indices (which were determined with reference to a homologous series of C5–C24 n-alkanes) by comparing their mass spectra with those stored in the National Institute of Standards and Technology (NIST) MS search 2.2 library. Agilent MassHunter Qualitative Analysis 10.0 data acquisition software was applied to analyze the GC–MS data. The amount of separated compounds was calculated in the peak areas using the normalization method without correction factors.

### 3.9. Determination of the Total Phenolic Content

The total phenolic content (TPC) in the *G. hederacea* extracts was determined using the Folin–Ciocalteu colorimetric method described by Kähkönen et al. [36], with slight modifications. In brief, 20 µL of extract was added to a 96-well plate and mixed with 100 µL of 10% Folin–Ciocalteu reagent, followed by the addition of 80 µL 7.5% Na_2_CO_3_ solution. After incubation at room temperature for 30 min in the dark with slight shaking, the absorbance at 765 nm was measured on a Hidex Sense microplate reader. Gallic acid was used as a standard for the calibration curve. The total phenolic content was expressed as mg of gallic acid equivalent (GAE) per g of lyophilized extract. All measurements were made in triplicate.

### 3.10. Determination of the Total Flavonoid Content

The total flavonoid content (TFC) was determined using the colorimetric method described by Wang et al. [37]. Briefly, 100 μL of sample solution was mixed with the same volume of 2% aluminum trichloride in methanol. Similarly, a blank was prepared by adding 100 μL of sample solution to methanol without AlCl_3_. After incubation at room temperature for 10 min, the absorbance at 415 nm was measured on a Hidex Sense microplate reader. The calibration curve was prepared using various concentrations of quercetin (0–250 μg/mL) dissolved in methanol. TFC was expressed as mg of quercetin equivalent (QE) per g of lyophilized extract. All measurements were made in triplicate.

### 3.11. 2,2-Diphenyl-1-Picrylhydrazyl (DPPH) Free Radical Scavenging Assay

DPPH was used to assess the free radical scavenging (antioxidant) properties of the *G. hederacea* extracts. DPPH radical scavenging activity was measured according to Brand-Williams et al. [38], with some modifications. For the assay, 20 μL of extract diluted in water was mixed with 180 μL of DPPH in methanol (40 μg/mL) in the wells of a 96-well plate. The plate was kept in the dark at room temperature for 15 min. Decreases in absorbance at 517 nm were measured using a Hidex Sense microplate reader. Ascorbic acid solutions in the concentration range of 0–800 μg/mL were used as a standard. The extract was tested in a range of concentrations to establish the IC50 (the concentration that reduced the absorbance of DPPH by 50%). The radical scavenging activity was calculated using the following formula:

DPPH radical scavenging activity % = [(A_0_–A_1_)/A_0_] × 100, where A_0_ is the absorbance of the control and A_1_ is the absorbance of the sample.

### 3.12. Isolation of Bone-Marrow-Derived Macrophages

Bone-marrow-derived macrophages (BMDMs) were isolated from male C57BL6/J inbred mice (18–20 weeks old, Envigo, Netherlands). BMDMs were cultured for 6–7 days in RPMI-1640 medium with GlutaMAX supplemented with 10% FBS, 1% antibiotics (100 U/mL penicillin and 100 μg/mL streptomycin), and 10 ng/mL M-CSF. The experimental procedures were carried out in accordance with the guidelines of the European Community (2010/63/EU) and local laws and policies, and were approved by the Latvian Animal Protection Ethical Committee, Food and Veterinary Service, Riga, Latvia.

The Petri dish containing BMDMs was washed twice with HBSS. The cells were detached with 0.5% trypsin and placed in RPMI-1640 medium with 10% FBS and 1% antibiotics, and the cell suspension was centrifuged at 300× *g* at room temperature for 5 min. The cells were resuspended in RPMI-1640 medium supplemented with 10% FBS and 1% antibiotic, and seeded in 12- or 96-well plates. The cells were then incubated in an incubator at 37 °C for at least 1 h prior to the experiment.

### 3.13. Evaluation of Cell Viability with the MTT Assay

The viability of BMDMs after 24 h of incubation with different concentrations of *G. hederacea* extract was determined using the MTT assay. BMDMs were seeded in 96-well plates at a final concentration of 50 × 10^4^ cells/mL. After incubation with the extract, the cells were further incubated with MTT solution (1 mg/mL) at 37 °C for 1–2 h. Subsequently, the medium was aspirated, and isopropanol was added to each well to dissolve the formazan crystals that formed during the incubation period. The absorbance was determined spectrophotometrically at 570 nm using a reference wavelength of 650 nm on a Hidex Sense microplate reader.

### 3.14. Treatment of Bone-Marrow-Derived Macrophages with Extract, Polarization toward the M1 Phenotype, and Analysis by Flow Cytometry

BMDMs were seeded at a density of 30 × 10^4^ cells/well in 1 mL of culture medium. The cells were stimulated with 5 ng/mL LPS and 10 U/mL murine IFN-γ for macrophage polarization toward the M1 (proinflammatory) phenotype, together with GH01 extracts (250 µg/mL and 500 µg/mL) or 100 µM of rosmarinic acid, chlorogenic acid, and caffeic acid for 24 h.

The cells were then washed twice with HBSS and harvested with 0.5% trypsin. Then, an RPMI medium with 10% FBS and 1% antibiotic was added, and the cell suspension was centrifuged at 300× *g* for 5 min. The cells were then incubated with specific conjugated antibody mixtures (at a dilution of 1:200) for 45 min on ice in the dark. For M1-polarized cells, the mixture contained the following monoclonal antibodies: FITC-conjugated anti-mouse F4/80, phycoerythrin (PE)-conjugated anti-mouse CD86, and biotin-conjugated anti-mouse CD80. After staining, the expression of the markers was analyzed by flow cytometry (BD FACSMelody^TM^, BD Biosciences, San Jose, CA, USA).

### 3.15. Statistical Analysis

Three biological replicates per sample were used for experiments with BMDMs, and two technical replicates were analyzed per biological replicate.

Quantitative results are expressed as the mean ± standard error of the mean (SEM) of three independent experiments and were analyzed using the computer software GraphPad Prism 8. Statistical analyses were conducted using one-way analysis of variance (ANOVA), followed by the Dunnett test. An unpaired t-test was used for the MTT assay. The results of the MTT test are expressed as the mean ± standard deviation (SD). Values of *p* < 0.05 were considered significant. A heatmap with scaled data was created in the R package pheatmap [39]. Principal component analysis (PCA) was performed using the FactoMineR [40] and factoextra [41] packages in R software version 4.1.4.

## 4. Conclusions

No detailed guidance on the cultivation of *G. hederacea* has been published thus far. Several factors, such as genotype, harvesting time, growing location, and other conditions, could affect the chemical composition of the plant and its biological activity [20]. Our study on *G. hederacea* demonstrated differences between the accessions tested. The wild populations of *G. hederacea* under field conditions differed significantly in their plant height, yield, and development. *G. hederacea* could be harvested several times during the season. The data obtained clearly revealed the presence of chemical variety within the accessions of *G. hederacea* and indicated the chemical compositions were different between harvesting periods. From the point of view of commercial cultivation and based on the different chemical compositions of the flowering and vegetative shoots of *G. hederacea*, the most suitable harvesting time for polyphenolic compounds is the flowering phase, while for essential oils, it is the vegetative phase. To harvest *G. hederacea* parts that are rich with caffeic acid, the vegetative phase is more appropriate. The changes in the content of flavonoids in *G. hederacea* plant extracts are more pronounced with changes in the time of plant collection rather than in the phenolic acid (chlorogenic acid, rosmarinic acid, and caffeic acid) content. *G. hederacea* can be successfully grown under organic farming conditions, and cultivation does not significantly affect its chemical composition or biological activity compared to wild-grown plants. Accession GH03 was the most suitable sample for commercial cultivation within the tested samples and cultivation seasons.

## Figures and Tables

**Figure 1 plants-11-00819-f001:**
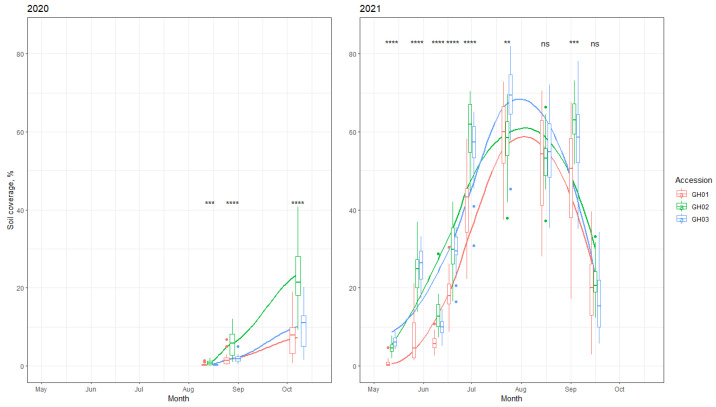
Development intensity (soil coverage %) of the tested *G. hederacea* accessions in 2020 and 2021. Boxplots represent the median (line), 25–75% quartiles (boxes), ranges (whiskers), and extreme values (cross). The symbols above the boxplots represent ANOVA *p* values within dates. ns—*p* > 0.05; **—*p* <= 0.01; ***—<= 0.001; ****—*p* <= 0.0001. The smoothed lines represent the loess function for each group.

**Figure 2 plants-11-00819-f002:**
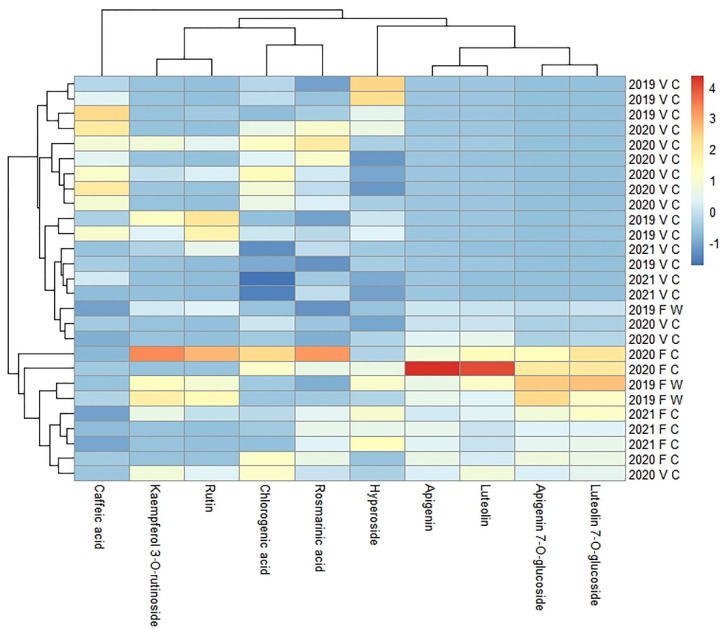
Variations in the chemical components of *G*. *hederacea* ethanol extracts from flowering and vegetative shoots. The legend denotes scaled values of the chemical constituents. F—flowering aerial part, V—vegetative aerial part, W—wild-grown, C—cultivated. The dendrogram represents a hierarchical clustering of the samples and ethanol extract constituents.

**Figure 3 plants-11-00819-f003:**
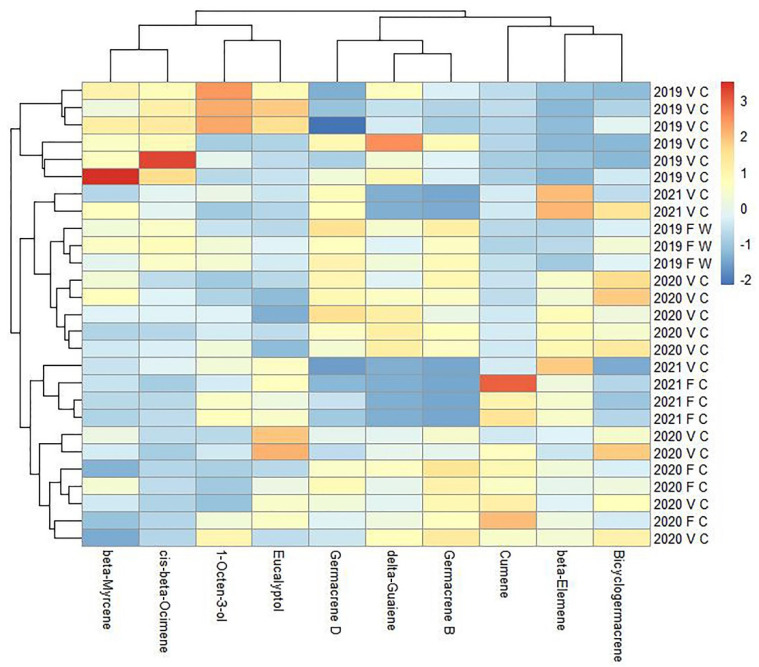
Variations in the chemical components of the essential oil of *G*. *hederacea* over three consecutive years. The legend denotes scaled values of the chemical constituents. F—flowering aerial part, V—vegetative aerial part, W—wild-grown, C—cultivated. The dendrogram represents a hierarchical clustering of samples and essential oil constituents.

**Figure 4 plants-11-00819-f004:**
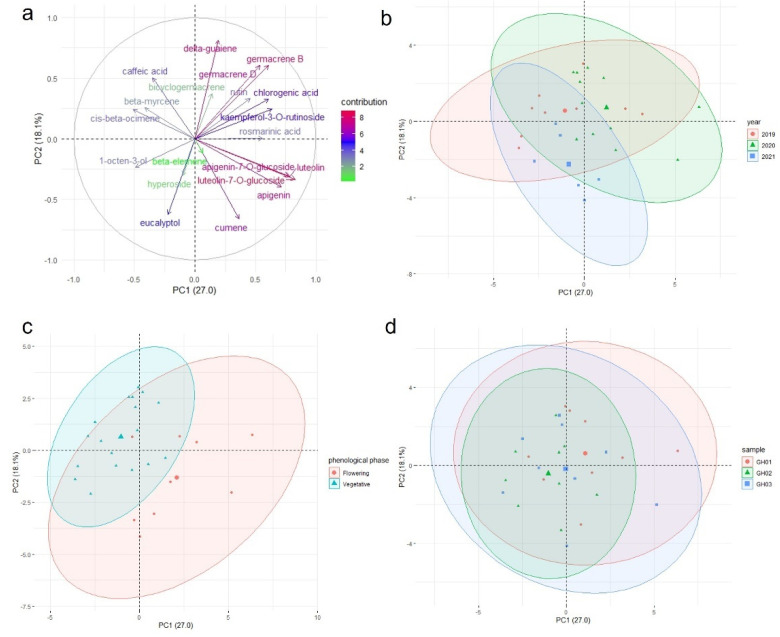
PCA loading plot (**a**) and scatter plots (**b**–**d**) of the compositions of essential oil and ethanol extracts of *G*. *hederacea*. The biplot vectors represent the direction and strength of the factor loading for the first two factors. Ellipses represent 95% confidence areas for the groups of two origins. Centroids are represented by the largest point of the same color.

**Figure 5 plants-11-00819-f005:**
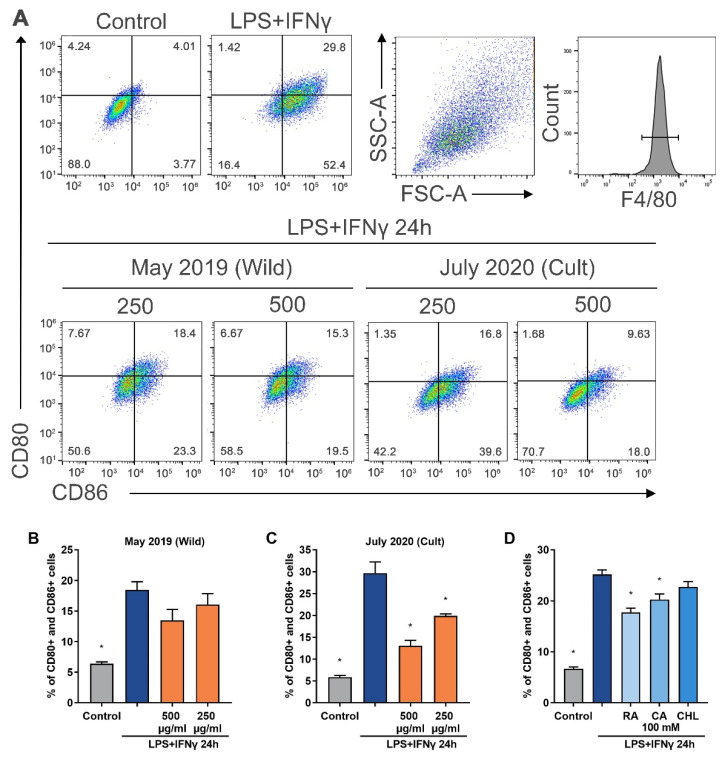
Flow cytometry analysis of bone-marrow-derived macrophage (BMDM) polarization toward the M1 phenotype. (**A**) Upper right quadrant: F4/80-positive cells were gated for double CD80 and CD86 analysis. The dot plot representation of a total of three independent experiments with two replicates is shown at the bottom of the figure. (**B**,**C**) Expression of the proinflammatory cell surface markers CD80 and CD86 was analyzed by flow cytometry 24 h after treating BMDMs with (GH01) extracts (250 µg/mL and 500 µg/mL) or (**D**) rosmarinic acid, chlorogenic acid, or caffeic acid (each 100 mM) and LPS/IFN-γ. Data are represented as the mean ± SEM of three independent measurements from two parallel experiments. Differences between the measurements were tested using one-way ANOVA followed by Dunnett’s test. * Significantly different from the LPS/IFN-γ control (*p* < 0.05).

**Figure 6 plants-11-00819-f006:**
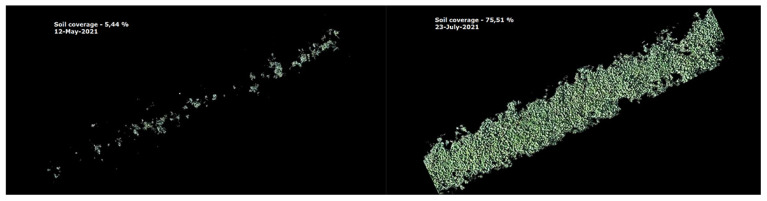
Soil coverage (%) for *G. hederacea* accession GH03 on 12 May and 23 July 2021.

**Table 1 plants-11-00819-t001:** Height of the plant and dry herb yield of the cultivated accessions (in different vegetation periods) of *G. hederacea*, 2021.

Sample	Plant Height, cm	Dry Herb Yield (12% Moisture) kg ha^−1^
Cultivated (flowering shoots, June 2021)		
GH01	13.5a	123.5a
GH02	19.0b	399.7b
GH03	20.2b	565.8b
Average:	17.6	
Cultivated (vegetative shoots, July 2021)		
GH01	11.6a	not harvested
GH02	7.9b	not harvested
GH03	10.8a	not harvested
Average:	10.1	-

Differences between the measurements were tested using two-way ANOVA. Different lowercase letters indicate the difference between the plant height and dry herb yield based on Duncan’s test (*p* < 0.05).

**Table 2 plants-11-00819-t002:** Phytocomponents identified in the 70% ethanol extracts of *G. hederacea*.

Peak	RT, min	Compound	Characteristic Ions ^1^, *m*/*z*		Calculated Elemental Composition ^2^	HRMS-MS/MS Fragments (ESI^+^), *m*/*z*	Parent Scan of Aglycone Fragment ^3^ (ESI^+^), Ion, *m*/*z*
			[M + H]^+^	[M − H]^−^			
1	6, 2	Chlorogenic acid isomer	355.103	353.085	C_16_H_18_O_9_	-	-
2	9, 0	**Chlorogenic acid**	355.103	353.085	C_16_H_18_O_9_	-	-
3	10, 0	Chlorogenic acid isomer	355.103	353.085	C_16_H_18_O_9_	-	-
4	11, 1	Quercetin 3-O-diglucoside	627.153	625.138	C_27_H_30_O_17_	465.100	627, 465 (303)
						303.049	
5	15, 1	Quercetin 3-O-galactosyl-rhamnoside	611.159	609.142	C_27_H_30_O_16_	449.108	611, 449 (303)
						303.050	
6	16, 5	Acylated quercetin diglycoside	755.202	753.185	C_33_H_38_O_20_	449.106	755, 449, (303)
						303.049	
7	16, 9	**Rutin**	611.157	609.143	C_27_H_30_O_16_	465.102	611, 465, (303)
						303.049	
8	17, 2	**Hyperoside**	465.102	463.085	C_21_H_20_O_12_	303.049	465 (303)
9	17, 6	**Luteolin 7-O-glucoside**	449.107	447.089	C_21_H_20_O_11_	287.054	449 (287)
10	18, 0	Acylated kaempferol diglycoside	739.207	737.190	C_33_H_38_O_19_	433.113	739, 433, (287)
						287.054	
11	18, 5	Acylated quercetin glycoside	609.145	607.127	C_27_H_28_O_16_	303.049	609 (303)
12	18, 5	**Kaempferol 3-O-rutinoside**	595.163	593.331	C_27_H_30_O_15_	-	595 (287)
13	19, 5	**Apigenin 7-O-glucoside**	433.113	431.096	C_21_H_20_O_10_	271.059	433 (271)
14	20, 1	**Rosmarinic acid**	-	359.075	C_18_H_16_O_8_	-	-
15	20, 3	Acylated kaempferol glycoside	593.146	591.133	C_27_H_28_O_15_	287.056	(287)
16	22, 7	**Luteolin**	287.054	285.040	C_15_H_10_O_6_	-	-
17	25, 2	**Apigenin**	271.058	269.045	C_15_H_10_O_5_	-	-

^1^ HRMS data, ^2^ mass difference within ± 5 mDa, ^3^
*m*/*z* of aglycone fragment in brackets, **bold**—identified by comparison to reference compounds.

**Table 3 plants-11-00819-t003:** Content of phenolic acids (µg/g of dry material) in 70% ethanol extracts of wild and cultivated (in different vegetation periods) *G. hederacea* accessions.

Sample	Chlorogenic Acid	Rosmarinic Acid	Caffeic Acid
Wild (flowering shoots, May 2019)			
GH01	3306.3	1318.5	122.7
GH02	3062.6	381.6	71.9
GH03	3214.4	2589.7	149.1
Average:	3194.4	1429.9	114.6
SD:	100.5	904.9	32.0
Cultivated (vegetative shoots, July 2019)			
GH01	2957.4	773.2	147.4
GH02	2501.0	379.5	137.7
GH03	3803.5	824.1	154.8
Average:	3087.3	658.9	146.6
SD:	539.6	198.7	7.0
Cultivated (vegetative shoots, August 2019)			
GH01	4312.1	3398.4	287.5
GH02	2859.2	2639.3	420.0
GH03	3991.2	2101.1	219.2
Average:	3720.8	2712.9	308.9
SD:	623.2	532.1	83.3
Cultivated (flowering shoots, May 2020)			
GH01	8418.5	12714.7	100.1
GH02	6270.3	5531.3	130.8
GH03	6240.2	5527.4	134.5
Average:	6976.3	7924.5	121.8
SD:	1019.8	3387.2	15.4
Cultivated (vegetative shoots, June 2020)			
GH01	6301.4	3812.4	124.9
GH02	4282.8	2392.2	132.9
GH03	3318.4	1265.5	92.7
Average:	4634.2	2490.1	116.9
SD:	1242.9	1042.1	17.4
Cultivated (vegetative shoots, July 2020)			
GH01	6332.9	9320.5	276.2
GH02	4721.0	7039.6	225.5
GH03	5316.5	6777.5	379.3
Average:	5456.8	7712.5	293.7
SD:	665.5	1142.0	64.0
Cultivated (vegetative shoots, September 2020)			
GH01	6629.7	4163.9	298.6
GH02	5867.4	3542.7	376.1
GH03	5429.9	4565.3	281.1
Average:	5975.7	4090.6	318.6
SD:	495.7	420.7	41.3
Cultivated (flowering shoots, June 2021)			
GH01	3864.6	5146.3	73.2
GH02	3361.7	5444.6	111.1
GH03	2944.8	4724.2	75.6
Average:	3390.4	5105.0	86.6
SD:	376.1	295.5	17.3
Cultivated (vegetative shoots, July 2021)			
GH01	1827.3	3570.2	119.4
GH02	1101.0	2534.1	195.3
GH03	1454.7	3548.6	111.5
Average:	1461.0	3217.6	142.1
SD:	296.5	483.4	37.8

The data representing each sample are the mean values of two measurements.

**Table 4 plants-11-00819-t004:** Content of flavonoids (µg/g of dry material) in 70% ethanol extracts of wild and cultivated (in different vegetation periods) *G. hederacea* accessions.

Sample	Apigenin 7-O-Glucoside	Kaempferol 3-O-Rutinoside	Rutin	Hyperoside	Luteolin-7 O-Glucoside	Apigenin	Luteolin
Wild (flowering shoots, May 2019)							
GH01	4598.1	760.2	4229.5	2990.8	3204.2	127.0	90.9
GH02	755.3	312.2	2665.0	1104.7	648.4	63.9	34.6
GH03	4334.8	958.7	5565.2	1463.9	1771.2	118.9	48.0
Average:	3229.4	677.1	4153.2	1853.1	1874.6	103.3	57.8
SD:	1752.8	270.4	1185.2	817.7	1046.0	28.0	24.0
Cultivated (vegeative shoots, July 2019)							
GH01	n.d	731.8	7174.1	1853.0	115.2	4.2	7.2
GH02	n.d	14.7	92.9	1284.1	45.9	n.d	5.3
GH03	n.d	22.4	190.8	4414.4	56.4	n.d	5.1
Average:	-	256.3	2485.9	2517.2	72.5	4.2	5.9
SD:	-	336.2	3315.3	1361.5	30.5	0.0	1.0
Cultivated (vegetative shoots, August 2019)							
GH01	n.d	363.4	6081.6	2129.6	72.6	n.d	7.1
GH02	n.d	26.6	633.7	2320.8	40.6	n.d	5.8
GH03	n.d	17.3	180.6	4296.9	48.3	n.d	5.9
Average:	-	135.8	2298.6	2915.8	53.8	-	6.3
SD:	-	161.0	2681.4	979.7	13.7	-	0.6
Cultivated (flowering shoots, May 2020)							
GH01	2833.2	1555.7	8805.6	1447.9	2660.3	150.0	104.0
GH02	2095.6	30.4	265.9	1131.4	1327.1	125.6	43.7
GH03	3705.3	30.8	321.4	2472.1	2561.6	524.6	234.7
Average:	2878.0	539.0	3131.0	1683.8	2183.0	266.7	127.5
SD:	657.9	719.0	4012.6	572.2	606.5	182.6	79.7
Cultivated (vegetative shoots, June 2020)							
GH01	1265.2	584.3	2847.4	1401.5	1111.9	85.7	76.8
GH02	360.0	13.1	135.6	617.1	333.6	59.3	33.2
GH03	411.8	16.2	162.8	1456.4	454.8	98.7	59.0
Average:	679.0	204.5	1048.6	1158.3	633.4	81.2	56.3
SD:	415.0	268.6	1272.0	383.4	341.9	16.4	17.9
Cultivated (vegetative shoots, July 2020)							
GH01	n.d	594.8	2923.6	1364.6	59.9	5.2	8.4
GH02	n.d	13.0	151.5	478.5	12.5	n.d	4.3
GH03	n.d	14.6	262.1	2529.9	27.5	n.d	6.2
Average:	-	207.4	1112.4	1457.7	33.3	-	6.3
SD:	-	273.9	1281.5	840.1	19.8	-	1.7
Cultivated (vegetative shoots, September 2020)							
GH01	n.d	230.9	2438.4	675.4	29.4	n.d	6.5
GH02	n.d	34.4	293.2	454.1	7.3	n.d	4.8
GH03	n.d	10.5	174.8	1303.0	25.2	n.d	6.2
Average:	-	92.0	968.8	810.8	20.6	-	5.8
SD:	-	98.7	1040.3	359.6	9.6	-	0.7
Cultivated (flowering shoots, June 2021)							
GH01	2174.6	493.2	1684.5	2879.5	1767.5	75.6	52.5
GH02	1385.3	9.2	167.8	2386.0	941.1	118.4	34.0
GH03	1496.1	11.9	191.6	3206.9	1172.4	97.0	32.0
Average:	1685.4	171.4	681.3	2824.1	1293.7	97.0	39.5
SD:	348.9	227.5	709.4	337.4	348.1	17.5	9.2
Cultivated (vegetative shoots, July 2021)							
GH01	n.d	142.0	3239.5	1251.7	18.0	n.d	n.d
GH02	n.d	6.6	100.5	707.9	10.9	n.d	n.d
GH03	n.d	6.6	80.2	547.4	14.6	n.d	n.d
Average:	-	51.7	1140.1	835.7	14.5	-	-
SD:	-	63.8	1484.6	301.4	2.9	-	-

n.d.—not detected. The data representing each sample are the mean values of two measurements.

**Table 5 plants-11-00819-t005:** Essential oil content (mL/kg ± SD of dry material) of wild and cultivated (in different vegetation periods) *G. hederacea* accessions.

Sample	Essential Oil, mL/kg ± SD
Wild (flowering shoots, May 2019)	
GH01	0.35 ± 0.02
GH02	0.52 ± 0.03
GH03	0.32 ± 0.02
Cultivated (vegetative shoots, July 2019)	
GH01	0.43 ± 0.03
GH02	0.41 ± 0.01
GH03	1.50 ± 0.05
Cultivated (vegetative shoots, August 2019)	
GH01	0.85 ± 0.03
GH02	0.76 ± 0.01
GH03	1.20 ± 0.04
Wild (flowering shoots, May 2020)	
GH01	0.75 ± 0.04
GH02	0.63 ± 0.02
GH03	2.98 ± 0.05
Cultivated (flowering shoots, May 2020)	
GH01	0.72 ± 0.02
GH02	0.75 ± 0.02
GH03	1.49 ± 0.03
Cultivated (vegetative shoots, June 2020)	
GH01	1.53 ± 0.05
GH02	0.75 ± 0.02
GH03	2.29 ± 0.04
Cultivated (vegetative shoots, July 2020)	
GH01	1.03 ± 0.04
GH02	0.77 ± 0.03
GH03	2.62 ± 0.05
Cultivated (vegetative shoots, September 2020)	
GH01	1.40 ± 0.03
GH02	1.36 ± 0.03
GH03	1.47 ± 0.03
Cultivated (flowering shoots, June 2021)	
GH01	0.71 ± 0.01
GH02	0.70 ± 0.01
GH03	0.73 ± 0.02
Cultivated (vegetative shoots, July 2021)	
GH01	1.80 ± 0.03
GH02	1.44 ± 0.03
GH03	2.50 ± 0.03

**Table 6 plants-11-00819-t006:** Total phenolic content, total flavonoid content, and DPPH free radical scavenging activity of wild-grown and cultivated (in different vegetation periods) *G. hederacea* extracts.

Plant Sample	TPC (mg GAE/g Lyophilized Extract wt) ^a^	TFC (mg QE/g Lyophilized Extract wt) ^b^	IC_50_ Value of DPPH Radical Scavenging Activity (μg/mL)
Wild (1st harvest, flowering shoots, May 2019)			
GH01	52.95	15.48	766
GH02	48.29	28.55	796
GH03	68.13	24.85	571
Average ± SEM	56.46 ± 2.44	22.96 ± 2.75	711 ± 41
Cultivated (1st harvest, flowering shoots, July 2019			
GH01	54.70	26.02	583
GH02	60.86	31.58	498
GH03	60.27	29.96	572
Average ± SEM	58.61 ± 0.80	29.19 ± 1.17	551 ± 15
Cultivated (2nd harvest, vegetative shoots, August 2019)			
GH01	89.17	19.83	446
GH02	76.63	26.76	366
GH03	72.77	32.85	450
Average ± SEM	79.52 ± 2.02	26.48 ± 2.66	437 ± 22
Cultivated (1st harvest, flowering shoots, May 2020)			
GH01	83.82	41.23	357
GH02	66.76	41.63	568
GH03	53.13	42.15	535
Average ± SEM	67.91 ± 3.62	41.67 ± 0.19	487 ± 38
Cultivated (2nd harvest, vegetative shoots, June 2020)			
GH01	56.69	22.74	520
GH02	73.03	32.03	501
GH03	48.10	23.58	619
Average ± SEM	59.27 ± 2.98	26.12 ± 2.10	547 ± 21
Cultivated (3rd harvest, vegetative shoots, July 2020)			
GH01	108.64	23.19	273
GH02	84.93	28.16	353
GH03	90.14	25.01	365
Average ± SEM	94.57 ± 2.93	25.45 ± 1.03	330 ± 17
Cultivated (4th harvest, vegetative shoots, September 2020)			
GH01	87.92	21.47	365
GH02	87.92	26.94	271
GH03	87.76	26.11	373
Average ± SEM	87.87 ± 0.02	24.84 ± 1.20	336 ± 19
Cultivated (1st harvest, flowering shoots, June 2021)			
GH01	71.87	24.94	458
GH02	62.25	31.67	487
GH03	63.51	24.02	491
Average ± SEM	65.88 ± 1.23	26.88 ± 1.70	479 ± 6
Cultivated (2nd harvest, vegetative shoots, July 2021)			
GH01	66.49	19.18	484
GH02	71.68	29.22	417
GH03	66.85	24.07	493
Average ± SEM	68.34 ± 0.68	24.16 ± 2.05	465 ± 14
Ascorbic acid	-	-	43 ± 1

The aerial parts of wild accessions (GH01, GH02, GH03) of *G. hederacea* originating from different regions of Latvia were analyzed. Data are represented as the mean ± SEM of three independent experiments carried out in duplicate. ^a^ The total phenolic content is expressed as the gallic acid equivalents per gram (mg GAE/g) of the lyophilized extract. ^b^ The total flavonoid content is expressed as the quercetin equivalents per gram (mg QE/g) of the lyophilized extract. The ^b^ IC_50_ (μg/mL) value corresponds to the concentration that can scavenge 50% of DPPH free radicals.

## Data Availability

Data available on request.

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
