# Peer review of "Wild-Grown and Cultivated Glechoma hederacea L.: Chemical Composition and Potential for Cultivation in Organic Farming Conditions"

_plants, 2022, doi:10.3390/plants11060819_

Round 1
Reviewer 1 Report
This contribution presents and discusses the potential for commercial production of G. hederacea and compared the chemical composition and activity of 19 aqueous ethanol extracts and steam-distilled essential oils from wild-grown and cultivated G. hederacea collected in different harvesting periods. The work has application potential due to the growing interest in use of medicinal plants well known in traditional medicine for its healfth promoting features, in that antiinflammatory, antibacterial, antiviral and anticancer properties.
Overall, I found this study very interesting and I have no many comments on the manuscript. My main concern is the methodological issue of sourcing raw material for testing. I believe that in order to draw conclusions about similarities or differences between wild and cultivated plants, it is necessary to compare raw materials that grew under identical soil and climatic conditions. Otherwise, the authors risk misinterpretation of the results obtained, which are greatly influenced by the environmental conditions of cultivation. The article does not include a Conclusion section, which according to the requirements for authors should be placed after the Material and methods section. I consider it a quite big mistake, which, however, can and should be corrected. Nevertheless, manuscript is quite good written, has a clear structure and brings interesting and valuable preliminary information.
Reviewer 2 Report
The manuscript entitled "Wild-grown and cultivated Glechoma hederacea: chemical composition and potential for cultivation in organic farming conditions" is complex, contains many well-interpreted results and is accompanied by representative figures. I would like to make a few remarks: - use of the term "aqueous ethanol extracts" - may need to be replaced - Could you explain the reasons for the passage of this medicinal plant in the culture? It may require a justification for this complex topic.
Reviewer 3 Report
I think it is one of the few studies on the composition identification and efficacy analysis of essential oils for traditional medicinal herbs in different growth periods. The writing of the manuscript and the planning of the experiments are presented above the standard. I believe that it can also bring a huge amount of information to the readers of related research fields. However, some parts of this manuscript still need to be revised before publication.
1.The abstract part should contain data on anti-inflammatory or antioxidant properties.
2.Please add the number of replicates for each experiment
Reviewer 4 Report
The paper entitled: "Wild-grown and cultivated Glechoma hederacea: chemical composition and potential for cultivation in organic farming conditions" takes into account a few investigated species, worthy to be considered for the possibility of a more standardized cutlivation and for further studies on biological activities.
The work is well performed and different techniques and approaches have been considered.
I recommend some minor revisions:
move table 1, 2 and 3 in the main text, they make results more sounding.
Add a comprehensive "conclusions" paragraph.
Please check botanical technicism: add L. or Linneus after the firts time you cite the species and correct the botanical family, not in italic.
Table s1 and s2 in the main text
conclusion
Reviewer 5 Report
The quality of manuscript titled Wild-grown and cultivated Glechoma hederacea: chemical composition and potential for cultivation in organic farming conditions is high and the part material and methods is clear. Hovever I have some remarks:
1) In introduction could be good to write more details about chemical composition and biological activity.
2) In manuscript I notice the lack of discussion (comparison the obtained results with those in the literature)
